# The Effects of Matcha and Decaffeinated Matcha on Learning, Memory and Proteomics of Hippocampus in Senescence-Accelerated (SAMP8) Mice

**DOI:** 10.3390/nu14061197

**Published:** 2022-03-11

**Authors:** Kiharu Igarashi, Makiko Takagi, Yoichi Fukushima

**Affiliations:** 1Faculty of Agriculture, Yamagata University, 1-23 Wakaba-machi, Tsuruoka, Yamagata 997-8555, Japan; 2Yamagata Prefectural Fishery Research Institute, 594 Okuzure, Kamo, Tsuruoka, Yamagata 997-1204, Japan; takagima@pref.yamagata.jp; 3Department of Health Food Sciences, University of Human Arts and Sciences, 1288 Magome, Iwatsuki-ku, Saitama-shi, Saitama 339-8539, Japan; qzc04363@gmail.com

**Keywords:** matcha green tea, decaffeinated matcha, SAMP8, cognitive impairment, proteomics, hippocampus protein

## Abstract

Although the benefits of the consumption of green tea and its components, including catechins and theanine, regarding aging, memory impairment and age-related cognitive decline have been investigated in senescence-accelerated prone mice (SAMP8), studies that simultaneously measured the kinds of proteins that vary in their expression due to the administration of green tea and its extracts were not found. In this study, the effect of dietary and decaffeinated matcha on protein expression in the hippocampus of SAMP 8 was examined comprehensively, mainly using proteomics. Although improvements in memory and the hair appearance of the back coat were limited upon administering the samples, the following regulations were observed in some of the proteins involved in neuron degeneration, Parkinson’s and Alzheimer’s diseases, synapse transmission and nerve cell plasticity, antioxidation, glutamate transport and metabolism, GABA (γ-amino butyric acid) formation and transport and excitatory amino acid transporters: proteins downregulated upon sample intake (*p* < 0.05): brain acid-soluble protein 1, microtubule-associated protein tau, synapsin-2, sodium- and chloride-dependent GABA transporter; proteins that tended to decrease upon sample intake (0.05 < *p* < 0.10): Parkinson’s disease (autosomal recessive and early-onset) 7 and synapsin-1; proteins upregulated upon sample intake (*p* > 0.95): glutathione S-transferase Mu 1, tubulin alpha-1A chain, dynamin-2, calcium/calmodulin-dependent protein kinase type II subunit gamma and tyrosine 3-monooxygenase/tyrosine 5-monooxygenase activation protein epsilon polypeptide; proteins that tended to increase upon sample intake (0.95 > *p* > 0.90): glutathione S-transferase Mu7 and soluble carrier family 1 (glial high-affinity glutamate transporter); proteins that tended to decrease: sodium- and chloride-dependent GABA transporter 3. These results indicate that matcha and decaffeinated matcha could reduce aging and cognitive impairment by regulating the expression of these proteins. Furthermore, these proteins could be used as markers for the evaluation of food and its available components for reducing aging and cognitive impairment.

## 1. Introduction

The number of people with dementia is now over 50–70 million worldwide, and it is presumed to increase by 10 million every year. As the quality of life in old age decreases through cognitive impairment, protection through nutrition and the avoidance of lifestyle-related diseases is essential to continue a healthy life and to save medical expenses. Reflecting these situations, studies concerning protection from the negative impacts of aging and cognitive impairment through the daily intake of certain foods and their components have been conducted by many researchers. As a result of these studies, protection from the negative impacts of aging and cognitive impairment by polyphenol and flavonoid intake [1,2]; improvements in cognitive functions using green tea and its components, such as tea catechins [3,4]; the prevention of learning and memory impairment in senescence-accelerated mice (SAMP8) using green tea catechin [5] and the attenuation of age-associated muscle loss through the administration of epigallocatechin-3 gallate contained in green tea [6] have been reported. However, further precise examination is needed to clarify the mechanisms of action of these compounds in the hippocampus [7]. Matcha is a traditional green tea from Japan, obtained by the preparation of green tea leaves (*Camellia sinensis*), which are cultivated under shade for approximately 3 weeks before harvest and processed into green tea fine powder and consumed in a thick suspension in hot water, delivering richer tea catechins and theanine, as well as fat-soluble compounds, such as lutein and vitamin K, which are not contained in green tea infusions. Recently, it was shown that matcha consumption improves cognition in human volunteers [4,8]. However, information on the underlying mechanism is still limited. Therefore, we tried to examine the improvement effect of matcha and decaffeinated matcha on cognitive impairment and their action mechanism mainly from the comprehensive analysis of hippocampus proteins by proteomics in SAMP8 mice.

To date, the influence of the consumption of green tea catechin on the expression of hippocampus proteins has mostly been examined by measuring its effects on the expression of a few brain (hippocampus) proteins, which were set as markers for aging and cognitive impairment in individual experiments [5], suggesting that an extensive analysis of proteins using proteomics might be necessary to confirm the proteins influenced by ingested samples and to examine their action mechanism. However, the influence of matcha intake on protein expression in the mouse hippocampus, which may be involved in improving aging and cognitive impairment, has never been examined extensively.

Thus, the influence of long-term matcha intake on protein expression and its mechanism was investigated mainly by proteomics in the hippocampus of SAMP8 mice.

Senescence-accelerated prone-8 (SAMP8) mice have been known to show age-related changes in learning and memory abilities [9,10,11], suggesting that SAMP8 is a valuable animal model for the investigation of the effect of food and its components on aging and cognitive impairment. Senescence-accelerated-resistant mice (SAMR1), which are reported to show a normal aging profile [12], were set as the control. Thus, in this study, we investigated the functions of matcha and decaffeinated matcha in SAMP8 mice.

In this study, first, the influence of the intake of matcha and decaffeinated matcha on appearance, such as glossiness of the back coat, coarseness of hair on the head and memory ability, was examined; then, hippocampus proteins influenced by the administered samples were mainly examined using proteomics to confirm the proteins influenced by administering samples, to examine their action mechanism and finally to find the benefits of matcha and decaffeinated matcha. Additionally, proteins showing differential expression between SAMP8 and SAMR1 through differential expression analysis were also examined to determine the proteins that could vary through aging and cognitive impairment and could be improved by matcha intake and decaffeinated matcha.

## 2. Materials and Methods

### 2.1. Preparation of Matcha and Decaffeinated Matcha

Matcha and decaffeinated matcha (low in caffeine content) were obtained from Kyoeiseicha Co., Ltd. (Osaka, Japan). Matcha is a green tea powder that has a high content of whole leaves and is favored to drink as a thick suspension. The matcha was produced as follows: first, specific green tea leaves harvested after cultivation in an environment without sunlight for many weeks in May (the first tea in the year) were steamed and dried to obtain tencha, followed by removing the leaf vein and grinding it to produce matcha powder. The decaffeinated matcha was prepared by washing tencha green tea leaves with hot water to remove caffeine, then producing matcha green tea powder [13].

### 2.2. Animal Care

The research was performed with permission from the Institutional Animal Care and Use Committee of Yamagata University (Permit number: R2139) and carried out according to the National Institutes of Health Guide for the Care and Use of Laboratory Animals and our guidelines at Yamagata University. Mice were kept individually in stainless steel cages with screen bottoms, with both their sides separated by a stainless-steel plate, under an environment with 12 h of lighting and off at 22 ± 2 °C and 40–60 humidity conditions. Abnormal behavior was not observed during feeding periods due to the use of these cages. All surgery was carried out under anesthesia using 2% isoflurane (Wako Pure Chemical Industries, Ltd., Osaka, Japan), and all efforts were made to minimize the pain inflicted on mice.

### 2.3. Animals and Diet

Eight-month-old senescence-accelerated mice (SAMP8, male) and senescence-accelerated-resistant mice (SAMR1, male) as the control (average weight, 25 g; SPF) were bought from the Shizuoka Laboratory Animal Center (SLC) (Hamamatsu, Shizuoka, Japan) and cared for according to the guidelines of Yamagata University for the care of experimental animals. After acclimating to the environment for three days, the mice were randomly divided into four groups of five or six mice and fed either a basal diet (SAMR1 = R1 and SAMP8 = P8 groups, n 6), a basal diet with matcha (SAMP8 + M = P8 + M group, n 6), or a basal diet with decaffeinated matcha (SAMP8 + L = P8 +L group, n 5). The diet composition of each experimental group is indicated in Table 1. The amounts of matcha and decaffeinated matcha were set to avoid abnormality in the behavior of mice in the preliminary experiment.

Food and water were taken freely for 98 days. Body weight and food intake were measured every four days. Learning and memory ability tests using the step-through test system (shock generator: MK-SG05, Muromachi Kikai Co., Ltd., Tokyo, Japan) were carried out from days 91 to 94 of the feeding. First, mice were placed in a light box, which was partitioned by an opener from a black box; then, the time needed to move from the lightbox to the black box was measured (first measurement). Shortly after, the mice moved to the dark box were exposed to electric stimulation at 0.08 mA for 0.3 s per mouse. Then, the mice were taken out of the black box and held for 30 s. The mice were placed again into the same light box as in the first measurement. This procedure was repeated thrice. The delays in the movement time to the black box from the light box were used to evaluate learning ability. After 3 measurements, mice were returned to the cage in which the mice were kept, then after one day were placed into the light box. The time required to move to the black box was used to measure memory ability (first measurement for memory ability test), followed by electrical stimulation at 0.8 mA for two seconds per mouse. Then, after three hours, memory ability was measured again (as the first measurement of the second memory ability test). After one and three days of the second memory test, the memory ability test was repeated without electric stimulation.

The method used to evaluate the degree of senescence in mice was in accordance with Hosokawa et al. [14]. Although Hosokawa et al. used 11 categories of age-associated changes in behavior and appearance and evaluated their degree of severity in grades, in this experiment, only the difference in the appearance, such as glossiness of the back coat, coarseness of hair on the head and loss of hair on the head (ranked by five grades, from 1 to 5, depending on the severity of symptom in this order) was scored, and the tentative mean of the grade per mouse was shown as an aging score.

On day 97 or 98 of the feeding period, half of the mice in each group were anesthetized using isoflurane 3–5 h after starving. After the collection of blood by heart puncture with a syringe, the brain was detached. The brain hippocampus was removed from the detached brain as soon as possible. The hippocampus was kept under –80 °C until analysis.

### 2.4. Measurements of Amino Acid and TBARS, and Proteomics

The frozen hippocampus was manually crushed and homogenized with cold 80% methanol (MeOH) using a plastic pestle, then ultra-sonicated for 15 min in an ice bath, followed by centrifugation at 11,000 rpm for 15 min to obtain the protein pellet and supernatant according to the method of Ivanisevic et al. [15]. The protein pellet was re-homogenized with cold 80% MeOH and then re-centrifuged. The same procedure was repeated thrice. The obtained protein pellet and supernatant were used for proteomics and analyses of amino acid and TBARS, respectively. TBARS was measured by the method of Uchiyama and Mihara [16], in which TBARS values were measured using the fluorometric method at Ex 532 and Em 560 nm. Then, the formed malondialdehyde (MDA)–TBARS complex was purified by transferal to an n-butyl alcohol layer. Amino acids were measured by the ninhydrin method using an amino acid analyzer (Hitachi L8900, Tokyo, Japan).

The protein pellet for proteomics was suspended in the RapiGest SF Surfactant (Waters, Milford, MA, USA) solution, then cysteine residue in protein was reduced with dithiothreitol and alkylated with iodoacetamide (Zhao et al., 2010) [17]. Then, it was digested with sequencing-grade modified trypsin (Promega, Madison, WI, USA) for 16 h at 37 °C. The digested sample solution, which was added with 10% TFA (0.5% TFA at the final concentration), was incubated at 37 °C for 30 min and centrifuged at 10,000× *g* for 20 min at 5 °C. The obtained supernatant was freeze-dried and dissolved with a small amount of Milli Q water and then centrifuged at 10,000× *g* for 20 min at 5 °C. Finally, the obtained supernatant was applied to a UPLC ESI-Q-TOF MS/MS system (Xevo Q-Tof MS, Waters, Manchester, UK).

The peptides in the supernatant were separated on the UPLC (nano AQUITY, Waters, Manchester, UK). The UPLC conditions were as follows: column BEH C-18 (i.d. 2.1 × 100 mm, 1.7 µm, Waters, Ireland, UK); eluent, 0.1% HCOOH (eluent A), 0.1% HCOOH in acetonitrile (eluent B); flow rate, 0.3 mL/min; column temperature, 30 °C; 0–100% B for 45 min with linear gradient. The spectra were obtained in the positive ion mode in micro Q-TOF with a mass range of 50–2000. The source temperature was 120 °C. Protein identification was carried out using Protein Lynx (Waters, Milford, MA, USA) and a database on the mice (taxonomy ID: 10088), which was provided by the National Center for Biotechnology Information (NCBI), Bethesda, MD, USA. The carbamidomethylating of cysteine and methionine oxidation was considered to be caused by variable modifications of tryptic peptides in MS/MS analysis. Differential protein expression analyses were performed after data normalization using the Protein Lynx TM Global Server (ver. 2.3) (Waters, Manchester, UK).

### 2.5. Statics

Data are expressed as means ± SEM. Six animals in each group, except for the SAMP8+L group (five animals), were used. Bartlett’s test verified the homogeneity of variance between treatments. Data were statistically analyzed using one-way analysis of variance for the measurements of the aging score, hippocampus (GABA/Glu) ratio, hippocampus TBARS and learning and memory scores. Tukey’s multiple range test was used for a post hoc analysis of significance. All comparisons were considered significant at *p* < 0.05. A significant difference in differential protein expression analysis, which was performed after data normalization using the Protein Lynx TM Global Server (ver. 2.3), was considered significant at *p* < 0.05 or *p* > 0.95.

## 3. Results

### 3.1. Body Weight, Aging Score (Appearance), Hippocampus (GABA/Glu) Ratio, Hippocampus and Serum TBARS Levels and Learning and Memory Scores

Diet intake and body weight gain tended to be small in senescence-accelerated-resistant mice (SAMR1) as the control (R1 group), compared with the SAMP8 fed with matcha (P8 + M group) and decaffeinated matcha (P8 + L group) or without (P8 group), especially at day 24 and after. Matcha and decaffeinated matcha-fed SAMP8 tended to score higher in both food intake and body weight (Figure 1).

Although the aging score did not differ significantly among groups, SAMP8 fed with matcha (P8 + M group) and decaffeinated matcha (P8 + L group) exhibited a lower tendency than those without any supplementation (P8). The hippocampus (GABA/Glu (glutamic acid)) ratio, which was expected to show a higher value in SAMP8 fed with a diet supplemented with antioxidative food [18], and the hippocampus TBARS value did not differ among the four groups (Figure 2).

Although learning and memory scores did not statistically vary among groups, SAMP8 without matcha and decaffeinated matcha (P8) feeding exhibited slightly higher values in learning scores than those fed both matcha and decaffeinated matcha (P8 + M and P8 + L) (Figure 3A). Memory scores taken 24 h after the first learning test tended to be higher in the SAMP8 fed with matcha (P8 + M) than in SAMP8 (P8), but this was not statistically significant (see score at 24 h in Figure 3B). Although the memory scores measured after additional electric shock tended to be higher in SAMP8 groups, irrespective of the supplementation of the matcha samples (P8 + M and P8+L) or not (P8) (see score at 24 h + l h in Figure 3B), those measured at the third time (24 h + 3 days) tended to be higher in SAMP8 fed with matcha (P8 + M) and decaffeinated matcha (P8 + L) than those of SAMP8 without these samples (P8) (see score at 24 h +3 days in Figure 3B).

### 3.2. Hippocampus Proteins and Differential Protein Expression

#### 3.2.1. Expression of Proteins Related to Neuron Degradation, Parkinson’s Disease and Alzheimer’s

The results for the expression of major proteins considered to be related to neuron degradation, Parkinson’s disease and Alzheimer’s are shown in Table 2. Proteins that exhibited significant fold changes in the expression of proteins in the P8 group compared with the R1 group, exhibited significant fold changes in the expression of proteins in the P8+M group compared with the P8 group, or exhibited significant fold changes in the expression of proteins in the P8 + L group compared with the P8 group, are listed in Table 2.

The effects of matcha and decaffeinated matcha on neuron degradation were tentatively examined from the amounts of brain acid-soluble protein-1 (accession: BASP1_MOUSE) as a marker of neuron degradation. A significant increase in brain acid-soluble protein in SAMP8, compared with SAMR1, was significantly suppressed by adding matcha and decaffeinated matcha to the basal diet (see the fold change value lower than one in P8 + M/P8 and P8 + L/P8). The fold change in brain acid-soluble protein in the P8 + L group against the P8 + M group exhibited a higher value than one, with a significant difference, indicating that the brain acid-soluble protein was higher in mice fed with decaffeinated matcha than matcha alone. Therefore, matcha might be more effective than decaffeinated matcha.

The fold change in the expression of Parkinson’s disease (autosomal recessive and early onset) 7 (accession: B2KFH8_MOUSE) and Parkinson’s disease (autosomal recessive and early onset) 7 (fragment) (accession: A2A816_MOUSE) in the P8 + M and P8 + L groups, compared with the P8 group, had a value lower than one, indicating that the expression of these proteins was reduced by feeding matcha and decaffeinated matcha to SAMP8. It is known that the expression of protein tau is increased in Alzheimer’s and Parkinson’s diseases [19,20]. In this experiment, the fold change in the expression of microtubule-associated protein tau (accession: TAU_MOUSE) in the P8 + M and P8 + L groups, compared with the P8 group, showed a value lower than one. This showed that the expression of microtubule-associated protein tau might be suppressed by feeding with matcha and decaffeinated matcha, and that the lower value of protein tau might be caused by the suppression of amyloid-β formation by matcha [21]. The fold change in the P8 + L group against the P8 + M group in brain acid-soluble protein tended to be higher than one, indicating that matcha may be more effective than decaffeinated matcha to protect against Parkinson’s and Alzheimer’s diseases.

#### 3.2.2. Expression of Proteins Involved in Synapse Transmission and Nerve Cell Plasticity

The results on the expression of proteins involved in synapse transmission and nerve cell plasticity in SAMP8 are shown in Table 3. The fold change in synapsin-2 (SYN2_MOUSE) in the P8 + M group, compared with the P8 group, exhibited values lower than one, and the expression was significantly higher in the P8 group than in the P8 + M and P8+L groups, indicating that the expression of synapsin-2 was suppressed by the feeding with matcha and decaffeinated matcha. Although the fold change in the expression of synapsin-2 in the P8 group, compared with the R1 group (P8/R1), was lower than one, its value was higher than that in the P8 + M/P8 and P8 + L/P8 ratio (comparison between 0.91 in P8/R1 and 0.85 and 0.77 in P8 + M/P8 and P8 + L/P8, respectively), indicating that the expression of synapsin-2 might be suppressed more strongly by feeding with matcha and decaffeinated matcha. The fold change in the expression of hippocampal synapsin-1 (accession: SYN1_MOUSE) in the P8 + M and P8 + L groups, compared with the P8 group, also exhibited values lower than one, showing that the expression of synapsin-1 also tended to be suppressed by feeding with matcha and decaffeinated matcha. Nevertheless, the fold change values in synapsin-1 tended to be higher than those in synapsin-2, suggesting that synapsin-2 might be influenced more than synapsin-1 by feeding with matcha and decaffeinated matcha.

The fold change in the expression of hippocampal tubulin alpha-1A (accession: TBA1A_MOUSE) in the P8+M and P8+L groups, in comparison with the P8 group, exhibited a value higher than one, with a statistically significant difference. These results indicate that the expression of hippocampal tubulin alpha-1A might be enhanced by feeding with matcha and decaffeinated matcha (Table 3).

The fold change in the expression of hippocampus dynamin (accession: G3X9G4_MOUSE) in the P8 + L group, compared with the P8 group, exhibited values higher than one and indicated a significant fold change, indicating that the expression of hippocampal dynamin could be enhanced through feeding with matcha and, mainly, decaffeinated matcha, and the fold change in the P8 + L group was significantly higher than that of the P8 + M group, indicating that decaffeinated matcha might be more effective in enhancing the expression of hippocampus dynamin (Table 3). The other dynamin with a different accession number also showed the same differentiation as that in the accession G3X9G4_MOUSE. Additionally, as dynamin is among the representative proteins concerning synaptic function and structure, the upregulation of these proteins was considered to be more important for preserving synaptic plasticity [5].

Since it is reported that the expression of hippocampus synapsin-2 in aged rats is higher than that in younger ones [22], that tubulin alpha decreases with age [22], and further that dynamin decreases with age [23], the opposite results in the expression of these proteins in this experiment might confirm the protective effects of matcha and decaffeinated matcha on aging.

#### 3.2.3. Expression of Hippocampus Antioxidative Enzymes

The results on the expression of hippocampal antioxidative enzymes in SAMP8 are shown in Table 4. The fold change in the expression of hippocampal peroxiredoxin 5, isoform CRA_a (accession: Q9JHL8_MOUSE), in the P8 + M and P8 + L groups, compared with the P8 group, exhibited values higher than one. The fold change in P8 + M/P8 was higher than that of P8 + L/P8, indicating that matcha might be more effective than decaffeinated matcha in increasing the expression of peroxiredoxin 5. A significant effect of feeding with matcha and decaffeinated matcha on the expression of peroxiredoxin other than peroxiredoxin-5 (accession: Q9JHL8_MOUSE) was not observed. (Table 4).

The fold change in the expression of hippocampal glutathione S-transferase Mu 1 (accession: A2AE89_MOUSE) in the P8 + M and P8 + L groups, compared with the P8 group, exhibited a value higher than one (1.36 and 1.35), and both increases in the fold change were significant (Table 4). These results indicate that the expression of hippocampal glutathione S-transferase was enhanced by feeding with matcha and decaffeinated matcha. The lack of fold change in the expression of these enzymes in the P8 group in comparison with the R1 group might indicate that the expression of these enzymes was hardly affected by the difference in the types of P8 and R1. However, the upregulation in the expression of hippocampal peroxiredoxin-5, isoform CRA_a and hippocampal glutathione S-transferase Mu 1 by matcha and decaffeinated matcha might show that these enzymes could be used as markers for the evaluation of aging and cognitive impairment.

#### 3.2.4. Expression of Hippocampal Proteins Involved in the Transmission of Glutamic Acid

The results on the expression of hippocampus proteins involved in the transmission of glutamic acid in SAMP8 are shown in Table 5. Fold changes in the expression of solute carrier family-1 (Glial high-affinity glutamate transporter), member 2 (accession: A2APL7_MOUSE), calcium/calmodulin-dependent protein kinase type II subunit gamma (accession: KCC2G_MOUSE) and tyrosine 3-monooxygenase/tryptophan 5-monooxygenase activation protein (accession: Q5SS40_MOUSE) in the P8 + M and P8 + L groups, in comparison with the P8 group, were higher than one. The expression was significantly higher or tended to be higher in the P8 + M and P8 + L groups, compared with the P8 group (Table 5), indicating that the expression of proteins involved in the transport and metabolism of glutamate [5] was upregulated by feeding with matcha and decaffeinated matcha. As the upregulation of calcium calmodulin-dependent protein kinase (CaMII) is effective for preventing memory decline and for the upregulation of synaptic plasticity-related protein in the hippocampus [20], it was considered that the upregulation of these proteins by matcha and decaffeinated matcha could play an essential role in preventing memory decline, aging and cognitive impairment. An increase in the expression of tyrosine 3-monooxygenase/tryptophan 5-monooxygenase activation protein by matcha and decaffeinated matcha showed that they might have enhanced GABA synthesis, which is considered useful in preventing cognitive impairment [24] (Table 5).

The fold change in the expression of sodium- and chloride-dependent GABA transporter 3 (accession: S6A11_MOUSE) in the P8 group, in comparison with the R1 group, was higher than one and tended to be statistically higher in P8 than R1, indicating that the expression of this protein was maintained at a higher level in P8 than R1. Alternatively, the fold change in the expression of sodium- and chloride-dependent GABA transporter-3 (accession: S6A11_MOUSE) in the P8 + M group, compared with the P8 group, was lower than one and exhibited a significant change, indicating that matcha might be able to effectively downregulate the expression of this protein. Nevertheless, significant downregulation by decaffeinated matcha was not observed.

#### 3.2.5. Expression of Proteins Involved in Cholinergic Neurostimulation Action and Excitatory Amino Acid Transporter

The results on the expression of proteins related to cholinergic neurostimulation action and excitatory amino acid transporters are shown in Table 6. The fold change in the hippocampal cholinergic neuro-stimulating peptide (accession: D3Z1V4_MOUSE) in the SAMP8 group, compared with the R1 group, and that in the P8 + M and P8 + L groups, compared with the P8 group, did not indicate any significant change (Table 6), suggesting that this protein was not expressed highly enough to alleviate neuronal disability in Alzheimer’s disease [25] or to clear glutamate released by neurons [26]. Alternatively, the expression of excitatory amino acid transporters, which is vital for the rapid removal of released glutamate from the synaptic cleft and for the termination of the postsynaptic actions of glutamate, was mostly unaltered at the statistical level, suggesting that the other glutamate transport in glial cells might be strongly related to the clearance of acidic amino acids from the synaptic cleft.

## 4. Discussion

As no difference in the aging score due to sample feeding was observed, breeding for a longer period or a higher dosage of the sample might have been necessary to clearly observe the difference in appearance, such as the aging score. The effects of matcha and decaffeinated matcha on learning and memory abilities, determined using the step-through test, loading 0.08 mA for 0.3 s per mouse (repeated thrice), was unclear in the first loading test; the memory ability tended to be improved in the second step-through test, which was performed through additional electric shocks of 0.8 mA for two seconds (repeated thrice) just after the first step-through test, suggesting that matcha and decaffeinated matcha may have the ability to improve memory ability, but this was not statistically significant. Stronger electric stimulation from the first electric shock without alternation in the latter stage might be necessary to obtain clearer results in the step-through test.

Although many reports have dealt with the physiological functions of green tea and its components, especially catechins on the protection of cognitive impairment [3,5], almost all of them were obtained by evaluating the effects of ingested samples on the expression of some proteins in the brain and hippocampus with learning and memory abilities in mouse models with cognitive impairments, such as SAMP8. To discover the protective effects of matcha and decaffeinated matcha against cognitive impairment, including their action mechanism, it is considered to be necessary to examine their effects comprehensively using methods such as proteomics, which makes it possible to examine the effects on the expression of many proteins holistically.

An outline of the results of the differential expression analyses between two groups are shown in Table 7 as increase/decrease arrows. Although a significant improvement in aging and memory abilities due to the ingestion of matcha and decaffeinated matcha was not observed, differential analyses and fold changes between the control and ingested groups in the expression of hippocampus proteins indicated that proteins mainly associated with neuron degeneration, Parkinson’s disease and Alzheimer’s (brain acid-soluble protein 1), glial fibrillary acidic protein, Parkinson’s disease (autosomal recessive and early onset) 7 and microtubule-associated protein tau are downregulated by matcha and decaffeinated matcha, and that proteins mainly associated with antioxidation (peroxiredoxin 5 and glutathione S-transferase Mu 1) are upregulated by matcha and decaffeinated matcha.

Proteins associated with synapse transmission and nerve cell plasticity, except for proteins of synapsin-2 and synapsin-1 (tubulin alpha-1A chain and dynamin-2), were upregulated. In contrast, synapsin-2 and synapsin-1 were downregulated. As it is reported that the amounts of synapsin-2 are higher in the hippocampus of older rats than in younger ones, that the amounts of tubulin alpha are lower in the hippocampus of old rats than young ones [22] and that dynamin is needed for memory formation [23], the opposite amounts of these proteins in the hippocampus in mice administered matcha and decaffeinated matcha indicated that the administered samples have the ability to ameliorate aging and cognitive impairment, presumably by increasing nerve cell plasticity. Proteins associated with glutamate transport and metabolism, and with GABA formation and transport, were upregulated, except for sodium- and chloride-dependent GABA transporter-3, suggesting that matcha and decaffeinated matcha may contribute to the prevention of aging and cognitive impairment by upregulating the expression of these proteins (solute carrier family 1 (glial high-affinity glutamate transporter) and calcium/calmodulin-dependent protein kinase type II subunit gamma) (Table 7). However, detailed mechanisms need to be examined in the future. In addition, it is necessary to examine the identification of effective components and their action mechanisms in the future.

Although solute carrier family-1 (glial high-affinity glutamate transporter) was upregulated by feeding with matcha and decaffeinated matcha, the expression of excitatory amino acid transporters was mostly not influenced by the ingestion of these samples, suggesting that glial glutamate transporters other than excitatory amino acid transporters may play an essential role in the clearance of glutamate in the synaptic cleft when mice were fed these samples. Although the administration of matcha and decaffeinated matcha influenced the expression of proteins involved in neuron degeneration, Parkinson’s disease and Alzheimer’s, antioxidation, synapse transmission and nerve cell plasticity and glutamate transport and metabolism, it is interesting to consider whether these influences were caused by the same compounds contained in the samples or not. These points remain to be solved in the future.

Some reports show that the long-term ingestion of caffeine partially improves hippocampus-dependent learning and memory abilities [27], indicating the possibility that one reason for the difference in protein expression between matcha- and decaffeinated matcha-fed mice might be caused by the difference in caffeine content. Since matcha and decaffeinated matcha contain large amounts of catechins, such as epigallocatechin (EGC), epigallocatechin gallate (EGCG), epicatechin (EC) and epicatechin gallate (ECG), which can ameliorate aging and cognitive impairment [28], these compounds may also be strongly related to the efficiency of matcha and decaffeinated matcha. However, as matcha and decaffeinated matcha also contain other compounds, such as theanine [29,30] and lutein [31], it may also be necessary to consider the effects of these compounds. It is known that theanine, which is abundant in matcha, exhibits preventive effects on stress-induced brain atrophy [32]; vitamin K and lutein, which are abundant in matcha, enhance angiogenic potential [33] and cortical capillary aging was prevented in matcha-fed mice [33], suggesting that studies administering individual compounds may be necessary to clarify active compounds.

As a conclusion of this experiment, it was found, mainly using proteomics of the hippocampus of SAMP8, that the ingestion of matcha and decaffeinated matcha could influence the expression of many proteins associated with memory, aging and cognitive impairment and could improve their expression, confirming the effect of matcha and decaffeinated matcha. Proteins showed differential expression upon the administration of matcha and decaffeinated matcha, which are expected to be used as markers for the evaluation of food and its components that show protective effects against aging and cognitive impairments.

## 5. Conclusions

The effects of the ingestion of matcha and decaffeinated matcha on aging, memory ability and the expression of hippocampus proteins that may be related cognitive impairment were investigated in a senescent-accelerated mouse model (SAMP8).

Although the improvement of memory ability by the ingestion of these samples was minor, the ingestion of these samples downregulated the expression of the hippocampus brain acid-soluble protein 1, microtubule-associated protein tau, syanpsin-2 and sodium- and chloride-dependent GABA transporter, and tended to downregulate the expression of Parkinson’s disease 7 and synapsin-1. Furthermore, it was shown that these samples upregulated the expression of hippocampus glutathione S-transferase Mu 1, tubulin alpha-1A chain, dynamin-2, calcium/calmodulin-dependent protein kinase type II subunit gamma and tyrosine 3-monooxygenase/tyrosine 5-monooxygenase activation protein epsilon polypeptide. These results indicate the possibility that matcha and decaffeinated matcha could reduce aging and cognitive impairment by regulating the expression of these proteins. Furthermore, these results suggest that these proteins could be used as markers for food evaluation and for the investigation of food components that are capable of reducing aging and cognitive impairment. In particular, we are the first to propose that the downregulation of synapsin-2 by matcha may be involved in the prevention of cognitive impairment.

## Figures and Tables

**Figure 1 nutrients-14-01197-f001:**
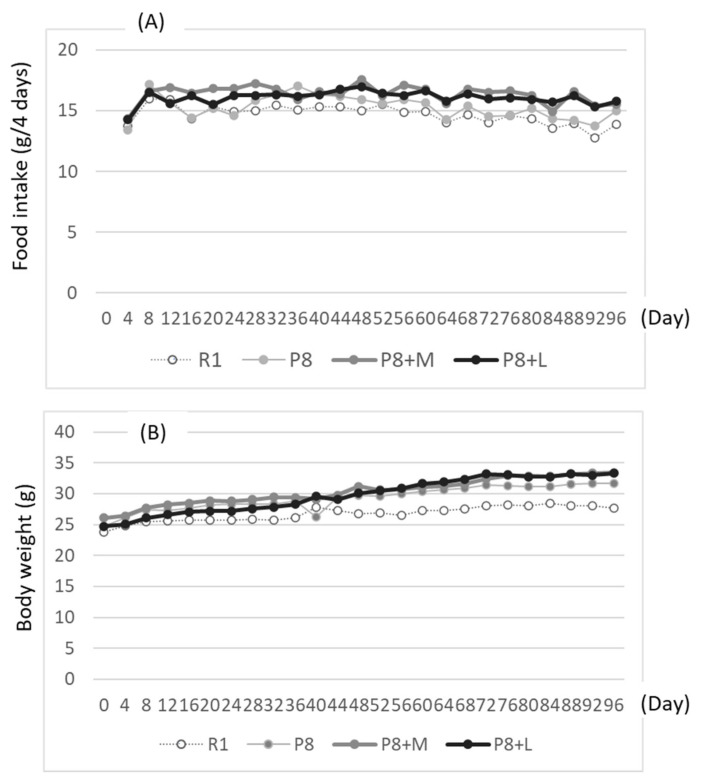
Effects of dietary matcha and decaffeinated matcha on food intake every 4 days (**A**) and body weight (**B**). R1: senescence-accelerated-resistant mice (SAMR1) group fed with basal diet; P8: senescence-accelerated mice (SAMP8) group fed with basal diet; P8 + M: SAMP8 group fed with both basal diet and matcha; P8 + L: SAMP8 group fed with both basal diet and decaffeinated matcha. Food intake and body weight every 4 days are means of each group.

**Figure 2 nutrients-14-01197-f002:**
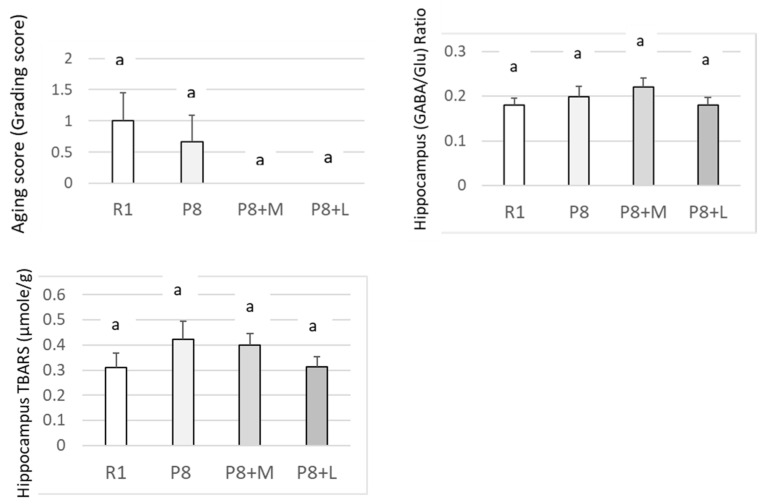
Effects of matcha and decaffeinated matcha on the aging score, hippocampus (GABA/Glu) ratio and hippocampus TBARS levels of SAMP8. R1: SAMR1 group fed with basal diet; P8: SAMP8 group fed with basal diet; P8 + M: SAMP8 group fed with both basal diet and matcha; P8 + L: SAMP8 group fed with both basal diet and decaffeinated matcha. Each value is mean ± SEM. *n =* 6 for each R1, P8 and P8 + M groups, and *n =* 5 for P8 + L group. Groups with the same alphameric character (a) do not differ statistically (*p* < 0.05). GABA, γ-aminobutyric acid; Glu, glutamic acid; TBARS, thiobarbituric acid reactive substances. Aging score was tentatively calculated by evaluating the degree of disappearance of glossiness of the back coat, coarseness of hair on the head and loss of hair on the head. Five grades (1–5: from weak to strong) were set depending on the magnitude of severity in appearance. Groups with a common letter did not differ statistically when analyzed by Tukey’s multiple comparison test (*p* < 0.05). Aging score was tentatively calculated by evaluating the degree of disappearance of glossiness of the back coat, coarseness of hair on the head and loss of hair on the head. Five grades (1–5: from weak to strong) were set depending on the magnitude of severity in appearance.

**Figure 3 nutrients-14-01197-f003:**
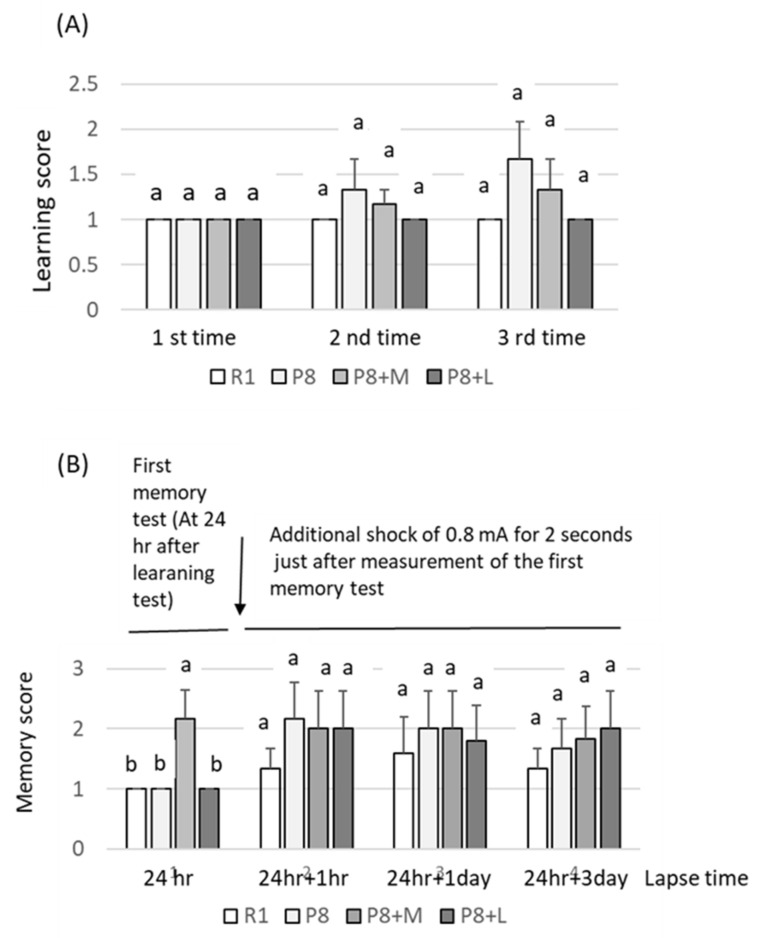
Effects of dietary matcha and decaffeinated matcha on learning (**A**) and memory (**B**) abilities. R1: SAMR1 group fed with basal diet; P8: SAMP8 group fed with basal diet; P8 + M: SAMP8 group fed with both basal diet and matcha; P8 + L: SAMP8 group fed with both basal diet and decaffeinated matcha. Learning and memory scores by the step-through test were defined as follows: score 1 in the case that mice placed in the light box moved to the dark box within 1 min; score 2 in the case that mice placed in the light box moved to the dark box after 1 to 2 min; score 3 in the case that mice placed in the light box took more than 2 min to move to the dark box. Groups with different alphanumeric characters (a or b) in groups grouped by number of times (Figure 3A) or elapsed time (Figure 3B), were significantly different when analyzed by Tukey’s multiple range comparison test (*p* < 0.05).

**Table 1 nutrients-14-01197-t001:** Composition of diets (%).

Experimental Groups	R1	P8	P8+M	P8+L
Mouse strains	SAMR1	SAMP8	SAMP8	SAMP8
Ingredient				
Casein	20	20	20	20
α-Cornstarch	55.95	55.95	55.45	55.45
Sucrose	7	7	7	7
Cellulose powder	5	5	5	5
Corn oil	7	7	7	7
Mineral mixture ^a^	3.5	3.5	3.5	3.5
Vitamin mixture ^b^	1	1	1	1
L-Cystine	0.3	0.3	0.3	0.3
Choline bitartarate	0.25	0.25	0.25	0.25
Matcha	0	0	0.5	0
Decaffeinated matcha	0	0	0	0.5
Total	100	100	100	100

R1 (n 6): senescence-accelerated-resistant mice (SAMR1) group fed with basal diet; P8 (n 6): senescence-accelerated-prone/8 (SAMP8) mice group fed with basal diet; P8 + M (n 6): SAMP8 group fed with both basal diet and matcha; P8 + L (n 5): SAMP8 group fed with both basal diet and decaffeinated matcha. Both ^a^ AIN-93G-MX and ^b^ AIN-93-VX were obtained from Clea Japan (Tokyo, Japan). Contents of epigallocatechin gallate (EGCG), epigallocatechin (EGC), epicatechin gallate (ECG) and epicatechin (EC) in hot 70% EtOH extracts of matcha and decaffeinated matcha were 185, 361, 21 and 63 mg, and 335, 475, 28 and 72 mg (mg/100 g powder), respectively, when determined by HPLC. Caffeine contents in hot H_2_O extracts of matcha and decaffeinated matcha, which were determined by HPLC, were 1.54 and 0.44 g (g/100 g powder), respectively.

**Table 2 nutrients-14-01197-t002:** Effects of feeding with matcha and decaffeinated matcha on neuron degeneration, Parkinson’s and Alzheimer’s diseases.

Accession	Description				
BASP1_MOUSE	Brain acid-soluble protein 1 OS = Mus musculus GN = Basp1 PE\ = 1 SV = 3				
		P8/R1	(P8 + M)/P8	(P8 + L)/P8	(P8 + L)/(P8 + M)
	Fold change (ratio)	1.31	0.63	0.85	1.28
	*p* value	1	0	0.04	0.99
GFAP_MOUSE	Glial fibrillary acidic protein OS = Mus musculus GN = Gfap PE = 1 SV = 4				
		P8/R1	(P8 + M)/P8	(P8 + L)/P8	(P8 + L)/(P8 + M)
	Fold change (ratio)	0.92	1.01	0.97	0.9
	*p* value	0.17	0.62	0.38	0.13
B2KFH8_MOUSE	Parkinson’s disease (autosomal recessive, early onset) 7 OS = Mus musculus GN = Park7 PE = 4 SV = 1				
		P8/R1	(P8 + M)/P8	(P8 + L)/P8	(P8 + L)/(P8 + M)
	Fold change (ratio)	1.01	0.33		
	*p* value	0.5	0.09		
A2A816_MOUSE	Parkinson’s disease (autosomal recessive, early onset) 7 (fragment) OS = Mus musculus GN = Park7 PE = 4 SV = 1				
		P8/R1	(P8 + M)/P8	(P8 + L)/P8	(P8 + L)/(P8 + M)
	Fold change (ratio)	1.01	0.35	0.46	0.93
	*p* value	0.52	0.17	0.09	0.43
B2KFI0_MOUSE	Parkinson’s disease (autosomal recessive, early onset) 7 (fragment) OS = Mus musculus GN = Park7 PE = 4 SV = 1				
		P8/R1	(P8 + M)/P8	(P8 + L)/P8	(P8 + L)/(P8 + M)
	Fold change (ratio)	1.14	0.7	0.74	0.89
	*p* value	0.86	0.2	0.18	0.4
TAU_MOUSE	Microtubule-associated protein tau OS = Mus musculus GN = Mapt PE = 1 SV = 3				
		P8/R1	(P8 + M)/P8	(P8 + L)/P8	(P8 + L)/(P8 + M)
	Fold change	1	0.81	0.67	1.06
	*p* value	0.47	0.01	0	0.92

R1 (n 6): SAMR1 group fed with basal diet; P8 (n 6): SAMP8 group fed with basal diet; P8 + M (n 6): SAMP8 group fed with both basal diet and matcha; P8 + L (n 5): SAMP8 group fed with both basal diet and decaffeinated matcha. Accession or number is a series of codes or numbers that are assigned to the identified protein by NCBI. Fold change is expressed as the ratio between two groups (numerator (NU) group/denominator (DE) group). *p* values higher than 0.95 indicate a possibility that the protein content is definitely higher in the numerator group than in the denominator group. *p* values lower than 0.05 indicate a possibility that the protein content is definitely higher in the denominator group than in the numerator group; OS, Organism species; GN, Geen (name); PE, Perionyx excavates, SV, splice variant.

**Table 3 nutrients-14-01197-t003:** Effects of feeding with matcha and decaffeinated matcha on the expression of proteins involved in synapse transmission and nerve cell plasticity.

Accession	Description				
SYN2_MOUSE	Synapsin-2 OS = Mus musculus GN = Syn2 PE = 1 SV = 2				
		P8/R1	(P8 + M)/P8	(P8 + L)/P8	(P8 + L)/(P8 + M)
	Fold change (ratio)	0.91	0.85	0.77	0.94
	*p* value	0.03	0	0	0.12
SYN1_MOUSE	Synapsin-1 OS = Mus musculus GN = Syn1 PE = 1 SV = 2				
		P8/R1	(P8 + M)/P8	(P8 + L)/P8	(P8 + L)/(P8 + M)
	Fold change (ratio)	0.95	0.93	0.93	1.01
	*p* value	0.13	0.1	0.08	0.6
O89052_MOUSE	Alpha-tubulin (fragment) OS = Mus musculus GN = Tuba1b PE = 2 SV = 1				
		P8/R1	(P8 + M)/P8	(P8 + L)/P8	(P8 + L)/(P8 + M)
	Fold change (ratio)	0.96	1.01	1.01	1.01
	*p* value	0.07	0.62	0.62	0.62
TBA1A_MOUSE	Tubulin alpha-1A chain OS = Mus musculus GN = Tuba1a PE = 1 SV = 1				
		P8/R1	(P8 + M)/P8	(P8 + L)/P8	(P8 + L)/(P8 + M)
	Fold change (ratio)	0.98	1.13	1.11	0.95
	*p* value	0.19	1	1	0
F8WIV5_MOUSE	Dynamin-2 OS = Mus musculus GN = Dnm2 PE = 3 SV = 1				
		P8/R1	(P8 + M)/P8	(P8 + L)/P8	(P8 + L)/(P8 + M)
	Fold change (ratio)	0.64	1.19	1.65	1.45
	*p* value	0	0.79	0.92	0.93
G3X9G4_MOUSE	Dynamin-2 OS = Mus musculus GN = Dnm2 PE = 3 SV = 1				
		P8/R1	(P8 + M)/P8	(P8 + L)/P8	(P8 + L)/(P8 + M)
	Fold change (ratio)	0.7	1.19	1.62	1.45
	*p* value	0.02	0.79	0.98	0.96
Q3TCR7_MOUSE	Dynamin-2 OS = Mus musculus GN = Dnm2 PE = 2 SV = 1				
		P8/R1	(P8 + M)/P8	(P8 + L)/P8	(P8 + L)/(P8 + M)
	Fold change (ratio)	0.7	1.11	1.57	1.4
	*p* value	0.04	0.66	0.95	0.97
Q3T9X3_MOUSE	Dynamin-2 OS = Mus musculus GN = Dnm2 PE = 2 SV = 1				
		P8/R1	(P8 + M)/P8	(P8 + L)/P8	(P8 + L)/(P8 + M)
	Fold change (ratio)	0.68	0.99	1.75	1.52
	*p* value	0.01	0.52	0.94	0.99

R1 (n 6): SAMR1 group fed with basal diet; P8 (n 6): SAMP8 group fed with basal diet; P8 + M (n 6): SAMP8 group fed with both basal diet and matcha; P8 + L (n 5): SAMP8 group fed with both basal diet and decaffeinated matcha. Accession or number is a series of codes or numbers that are assigned to the identified protein by NCBI. Fold change is expressed as the ratio between two groups (numerator (NU) group/denominator (DE) group). *p* values higher than 0.95 indicate a possibility that the protein content is definitely higher in the numerator group than in the denominator group. *p* values lower than 0.05 indicate a possibility that the protein content is definitely higher in the denominator group than in the numerator group.

**Table 4 nutrients-14-01197-t004:** Effects of feeding with matcha and decaffeinated matcha on the expression of proteins involved in antioxidative enzymes.

Accession	Description					
Q9JHL8_MOUSE	Peroxiredoxin 5, isoform CRA_a OS = Mus musculus GN = Prdx5 PE = 2 SV = 1					
		P8/R1	(P8 + M)/P8	(P8 + L)/P8		(P8 + L)/(P8 + M)
	Fold change (ratio)	0.83	1.36	1.22		0.87
	*p* value	0.13	0.96	0.8		0.1
PRDX5_MOUSE	Peroxiredoxin-5, mitochondrial OS = Mus musculus GN = Prdx5 PE = 1 SV = 2					
		P8/R1	(P8 + M)/P8	(P8 + L)/P8		(P8 + L)/(P8 + M)
	Fold change (ratio)	1.06	0.96	1		0.97
	*p* value	0.9	0.26	0.49		0.27
Q6GT24_MOUSE	Peroxiredoxin 6 OS = Mus musculus GN = Prdx6 PE = 2 SV = 1					
		P8/R1	(P8 + M)/P8	(P8 + L)/P8		(P8 + L)/(P8 + M)
	Fold change (ratio)	0.99	0.83	0.98		1
	*p* value	0.42	0.17	0.44		0.5
Q53ZU7_MOUSE	Peroxiredoxin 6 OS = Mus musculus GN = Prdx6 PE = 2 SV = 1					
		P8/R1	(P8 + M)/P8	(P8 + L)/P8		(P8 + L)/(P8 + M)
	Fold change (ratio)	1.13	0.84	0.94		1.14
	*p* value	0.66	0.19	0.4		0.75
E9QAC8_MOUSE	Glutathione S-transferase Mu 7 (fragment) OS = Mus musculus GN = Gstm7 PE = 4 SV = 1					
		P8/R1	(P8 + M)/P8	(P8 + L)/P8		(P8 + L)/(P8 + M)
	Fold change (ratio)	1.02	1.28	1.22		0.98
	*p* value	0.55	0.92	0.84		0.48
A2AE89_MOUSE	Glutathione S-transferase Mu 1 (fragment) OS = Mus musculus GN = Gstm1 PE = 3 SV = 1					
		P8/R1	(P8 + M)/P8	(P8 + L)/P8		(P8 + L)/(P8 + M)
	Fold change (ratio)	0.98	1.36	1.35		0.97
	*p* value	0.42	0.92	0.96		0.48

R1 (n 6): SAMR1 group fed with basal diet; P8 (n 6): SAMP8 group fed with basal diet; P8 + M (n 6): SAMP8 group fed with both basal diet and matcha; P8 + L (n 5): SAMP8 group fed with both basal diet and decaffeinated matcha. Accession or number is a series of codes or numbers that are assigned to the identified protein by NCBI. Fold change is expressed as the ratio between two groups (numerator (NU) group/denominator (DE) group). *p* values higher than 0.95 indicate a possibility that the protein content is definitely higher in the numerator group than in the denominator group. *p* values lower than 0.05 indicate a possibility that the protein content is definitely higher in the denominator group than in the numerator group.

**Table 5 nutrients-14-01197-t005:** Effects of feeding with matcha and decaffeinated matcha on the expression of proteins involved in glutamate transport and metabolism and in GABA formation and transport.

Accession	Description				
A2APL7_MOUSE	Solute carrier family 1 (glial high-affinity glutamate transporter), member 2 OS = Mus musculus GN = Slc1a2 PE = 2 SV = 1				
		P8/R1	(P8 + M)/P8	(P8 + L)/P8	(P8 + L)/(P8 + M)
	Fold change (ratio)	1.09	1.36	1.54	1.07
	*p* value	0.63	0.94	0.93	0.62
A2APL8_MOUSE	Solute carrier family 1 (glial high-affinity glutamate transporter), member 2 OS = Mus musculus GN = Slc1a2 PE = 4 SV = 1				
		P8/R1	(P8 + M)/P8	(P8 + L)/P8	(P8 + L)/(P8 + M)
	Fold change (ratio)	0.93	0.81	0.96	1.19
	*p* value	0.34	0.15	0.37	0.95
S6A11_MOUSE	Sodium- and chloride-dependent GABA transporter 3 OS = Mus musculus GN = Slc6a11 PE = 1 SV = 2				
		P8/R1	(P8 + M)/P8	(P8 + L)/P8	(P8 + L)/(P8 + M)
	Fold change (ratio)	1.14	0.76	0.9	1.07
	*p* value	0.93	0	0.25	0.67
Q3TY93_MOUSE	Calcium/calmodulin-dependent protein kinase type II subunit beta OS = Mus musculus GN = Camk2b PE = 2 SV = 1				
		P8/R1	(P8 + M)/P8	(P8 + L)/P8	(P8 + L)/(P8 + M)
	Fold change (ratio)	0.85	1.11	1.34	1.22
	*p* value	0	0.89	1	0.99
KCC2G_MOUSE	Calcium/calmodulin-dependent protein kinase type II subunit gamma OS = Mus musculus GN = Camk2g PE = 1 SV = 1				
		P8/R1	(P8 + M)/P8	(P8 + L)/P8	(P8 + L)/(P8 + M)
	Fold change (ratio)	0.95	1.2	1.48	1.23
	*p* value	0.3	0.92	1	0.95
A2A5N2_MOUSE	Tyrosine 3-monooxygenase/tryptophan 5-monooxygenase activation protein, beta OS = Mus musculus GN = Ywhab				
		P8/R1	(P8 + M)/P8	(P8 + L)/P8	(P8 + L)/(P8 + M)
	Fold change (ratio)	0.86	1.25	1.09	0.93
	*p* value	0.17	0.95	0.78	0.22
A2ACM8_MOUSE	Tyrosine 3-monooxygenase/tryptophan 5-monooxygenase activation protein, epsilon polypeptide (fragment) OS = Mus musculus GN = Ywhae PE = 4 SV = 1				
		P8/R1	(P8 + M)/P8	(P8 + L)/P8	(P8 + L)/(P8 + M)
	Fold change (ratio)	1.03	1.11	1.05	0.96
	*p* value	0.57	0.75	0.66	0.25
Q5SS40_MOUSE	Tyrosine 3-monooxygenase/tryptophan 5-monooxygenase activation protein, epsilon polypeptide				
		P8/R1	(P8 + M)/P8	(P8 + L)/P8	(P8 + L)/(P8 + M)
	Fold change (ratio)	1.04	1.32	1.13	0.9
	*p* value	0.69	0.99	0.91	0.07

R1 (n 6): SAMR1 group fed with basal diet; P8 (n 6): SAMP8 group fed with basal diet; P8 + M (n 6): SAMP8 group fed with both basal diet and matcha; P8 + L (n 5): SAMP8 group fed with both basal diet and decaffeinated matcha. Accession or number is a series of codes or numbers that are assigned to the identified protein by NCBI. Fold change is expressed as the ratio between two groups (numerator (NU) group/denominator (DE) group). *p* values higher than 0.95 indicate a possibility that the protein content is definitely higher in the numerator group than in the denominator group. *p* values lower than 0.05 indicate a possibility that the protein content is definitely higher in the denominator group than in the numerator group.

**Table 6 nutrients-14-01197-t006:** Effects of feeding with matcha and decaffeinated matcha on the expression of proteins involved in cholinergic neurostimulating action and on excitatory amino acid transporters.

●Proteins Related to Cholinergic Neurostimulating Action					
Accession	Description				
D3Z1V4_MOUSE	Hippocampal cholinergic neurostimulating peptide OS = Mus musculus GN = Pebp1 PE = 4 SV = 1				
		P8/R1	(P8 + M)/P8	(P8 + L)/P8	(P8 + L)/(P8 + M)
	Fold change (ratio)	0.94	1.06	0.97	0.95
	*p* value	0.27	0.67	0.38	0.42
D6RHS6_MOUSE	Hippocampal cholinergic neurostimulating peptide OS = Mus musculus GN = Pebp1 PE = 4 SV = 1				
		P8/R1	(P8 + M)/P8	(P8 + L)/P8	(P8 + L)/(P8 + M)
	Fold change (ratio)	0.94	1.12	1.07	1.01
	*p* value	0.33	0.75	0.71	0.57
●Excitatory amino acid transporter					
Accession	Description				
E9PVV4_MOUSE	Excitatory amino acid transporter 2 OS = Mus musculus GN = Slc1a2 PE = 4 SV = 1				
		P8/R1	(P8 + M)/P8	(P8 + L)/P8	(P8 + L)/(P8 + M)
	Fold change (ratio)	0.9	0.82	0.96	1.22
	*p* value	0.19	0.12	0.45	0.96
EAA2_MOUSE	Excitatory amino acid transporter 2 OS = Mus musculus GN = Slc1a2 PE = 1 SV = 1				
		P8/R1	(P8 + M)/P8	(P8 + L)/P8	(P8 + L)/(P8 + M)
	Fold change (ratio)	0.93	0.82	0.94	1.21
	*p* value	0.32	0.15	0.4	0.96

R1 (n 6): SAMR1 group fed with basal diet; P8 (n 6): SAMP8 group fed with basal diet; P8 + M (n 6): SAMP8 group fed with both basal diet and matcha; P8 + L (n 5): SAMP8 group fed with both basal diet and decaffeinated matcha. Accession or number is a series of codes or numbers that are assigned to the identified protein by NCBI. Fold change is expressed as the ratio between two groups (numerator (NU) group/denominator (DE) group). *p* values higher than 0.95 indicate a possibility that the protein content is definitely higher in the numerator group than in the denominator group. *p* values lower than 0.05 indicate a possibility that the protein content is definitely higher in the denominator group than in the numerator group.

**Table 7 nutrients-14-01197-t007:** Altered hippocampus proteins in SAMP8 compared with SAMR1 and the effects of matcha and decaffeinated matcha.

		Expression in SAMP8, Compared with SAMR1	Expression in SAMP8 Fed with Matcha, Compared with SAMP8	Expression in SAMP8 Fed with Decaffeinated Matcha, Compared with SAMP8
		(P8/R1)	(P8 + M/P8)	(P8 + L/P8)
Accession	●Neuron Degeneration, and Parkinson’s Disease and Alzheimer’s-Related Proteins	Expression Change	Expression Change	Expression Change
BASP1_MOUSE	Brain acid-soluble protein 1 OS = Mus musculus GN = Basp1 PE = 1 SV = 3	↑	↓	↓
B2KFH8_MOUSE	Parkinson’s disease (autosomal recessive, early onset) 7 OS = Mus musculus GN = Park7 PE = 4 SV = 1	→	↘	
A2A816_MOUSE	Parkinson’s disease (autosomal recessive, early onset) 7 (fragment) OS = Mus musculus GN = Park7 PE = 4 SV = 1	→	→	↘
B2KFI0_MOUSE	Parkinson’s disease (autosomal recessive, early onset) 7 (fragment) OS = Mus musculus GN = Park7 PE = 4 SV = 1	→		
TAU_MOUSE	Microtubule-associated protein tau OS = Mus musculus GN = Mapt PE = 1 SV = 3	→	↓	↓
Accession	●Antioxidative enzymes	Expression change Expression change Expression change		
Q9JHL8_MOUSE	Peroxiredoxin 5, isoform CRA_a OS = Mus musculus GN = Prdx5 PE = 2 SV =1	→	↑	→
E9QAC8_MOUSE	Glutathione S-transferase Mu 7 (fragment) OS = Mus musculus GN = Gstm7 PE = 4 SV = 1	→	↗	→
A2AE89_MOUSE	Glutathione S-transferase Mu 1 (fragment) OS = Mus musculus GN = Gstm1 PE = 3 SV = 1	→	↗	↑
Accession	●Proteins related to synapse transmission and nerve cell plasticity			
SYN2_MOUSE	Synapsin-2 OS = Mus musculus GN = Syn2 PE = 1 SV = 2	↓	↓	↓
SYN1_MOUSE	Synapsin-1 OS = Mus musculus GN = Syn1 PE = 1 SV = 2	→	↘	↘
TBA1A_MOUSE	Tubulin alpha-1A chain OS = Mus musculus GN = Tuba1a PE = 1 SV = 1	→	↑	↑
G3X9G4_MOUSE	Tubulin alpha-1A chain OS = Mus musculus GN = Dnm2 PE = 3 SV = 1	↓	→	↑
Q3TCR7_MOUSE	Dynamin-2 OS = Mus musculus GN = Dnm2 PE = 2 SV = 1	↓	→	↑
Q3T9X3_MOUSE	Dynamin-2 OS = Mus musculus GN = Dnm2 PE = 2 SV = 1	↓	→	↗
Accession	●Proteins related to glutamate transport and metabolism and to GABA formation and transport			
A2APL7_MOUSE	Solute carrier family 1 (glial high-affinity glutamate transporter), member 2 OS =Mus musculus GN = Slc1a2 PE = 2 SV = 1	→	↗	↗
S6A11_MOUSE	Sodium- and chloride-dependent GABA transporter 3 OS = Mus musculus GN = Slc6a11 PE = 1 SV = 2	↗	↓	→
Q3TY93_MOUSE	Calcium/calmodulin-dependent protein kinase type II subunit beta OS = Mus musculus GN = Camk2b PE = 2 SV=1	↓	→	↑
KCC2G_MOUSE	Calcium/calmodulin-dependent protein kinase type II subunit gamma OS = Mus musculus GN = Camk2g PE = 1 SV = 1	→	↗	↑
Q5SS40_MOUSE	Tyrosine 3-monooxygenase/tryptophan-5-monooxygenase activation protein epsilon polypeptide,	→	↑	↗
A2A5H2_MOUSE	Tyrosine 3-monooxygenase/tryptophan 5-monooxygenase activation protein, beta OS = Mus musculus GN = Ywhab PE = 2 SV = 1	→	↑	→
Accession	●Excitatory amino acid transporter			
E9PVV4_MOUSE	Excitatory amino acid transporter 2 OS = Mus musculus GN = Slc1a2 PE = 4 SV = 1	→	→	→
EAA2_MOUSE	Excitatory amino acid transporter 2 OS = Mus musculus GN = Slc1a2 PE = 1 SV = 1	→	→	→

The arrows ↓, ↑ and → show that protein expression in the numerator group in comparison with the denominator group was down- and upregulated and was not altered, respectively. The arrows ↗ and ↘ show increase and decrease tendencies, respectively. Number of mice in R1, P8, P8 + M and P8 + L groups are 6, 6, 6 and 5, respectively.

## Data Availability

All the data obtained by the study are available from the corresponding author on reasonable request.

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
