# Peer review of "The Effects of Matcha and Decaffeinated Matcha on Learning, Memory and Proteomics of Hippocampus in Senescence-Accelerated (SAMP8) Mice"

_nutrients, 2022, doi:10.3390/nu14061197_

Round 1

Reviewer 1 Report

Dear Authors,

Firstly I would like to congratulate you on an attempt to discuss such an important topic and to present the findings. I had a pleasure in reading this manuscript and please find below some of my comments/suggestions provided in Major section. I sincerely hope that this comments/suggestions assist in the improvements of the manuscript.

Major:

  1. I suggest that authors review the manuscript for use of academic expressions (in places). The meaning of some sentences can be difficult to understand by the reader (i.e. in several places there are short one line statements that could be incorporated with the following sentence). In saying this, the article is comprehensive and understandable and my comment is more orientated towards the reader of the manuscript.
  2. The title of the manuscript appears to be very convoluted. I suggest that author consider simplifying the title to more accurately represent the conducted study; (if I may suggest, along the lines of) “The effects of Matcha and decaffeinated Matcha on learning, memory and proteomics of hippocampus in senescence-accelerated (SAMP8) mice.” Please use this as a guide and modify this as you believe fits well.
  3. Abstract will require restructure as results included in this section are not adequately represented (p values missing and essential information). I suggest that authors restructure and cluster some of the proteins based on their functionality rather than up- or down-regulation.
  4. In introduction section , please provide adequate references for some statements in particular relating to the cognition, dementia and future trajectories. Also, the findings from animal and in vitro findings cannot be easily transferable into human clinical trials or prospective cohort studies. I suggest that authors focus on the animal trials findings in relation to the cognition, learning and proteomic analysis more.
  5. The composition of potentially active ingredients in the Matcha samples is beyond the theanine and EGCG. Please include more details on some of the bioactive compounds in Matcha samples and also their increase/decrease during the processing (decaffeination).
  6. Did authors analyze the composition of Matcha samples (any form of quantification – HPLC preferable). It is well established that catechin and theanine composition change during the processing.
  7. Methods section will also require simplification and clarification. For example, tests performed in triplicate and methodological approaches to mouse treatments will require restructure and better expressions of performed tasks.
  8. Figures will require simplification and definite inclusion of higher resolution. Consider providing two/three joint figures that enrapture better information presented. I find it rather difficult to observe any significant changes (if any).
  9. If the hypothesized findings were not observed (in particular using the electric shock) it is not appropriate to increase the voltage (or amperage) of the standardized stimuli specifically to obtain the desired response? Please rephrase and restructure first paragraph of the discussion.
  10. I would also suggest that authors consider interpretation of results between the parameters of the probability value. If something is less than 0.05, this should be non-significant (no changes observed). Please check the interpretation of results in the discussion section in several places.

Author Response

Thank you for your comments. Please find author's reply in the attachment.

Reviewer 2 Report

(1) Please comment on the animal’s well-being, did you monitor them for stress/discomfort within the cage?

(2) In this experimental, there were no significant differences in the aging score and cognitive impairment among groups. Please describe possible reasons or  limitations.

(3) Please describe the dosage and usage considerations for this experimental sample.

(4) The authors describe that the differential analyses and fold changes between control and ingested groups in the expression of hippocampus proteins. Please provide possible mechanisms or information on these outcomes.

Author Response

(The authors gave the same response as above.)

Round 2

Reviewer 1 Report

Dear Authors,

Congratulations on successfully addressing my comments and suggestions. Apart from checking the manuscript in several places for adequate grammar and use of academic expressions, I think this manuscript is well composed.

Author Response

Dear

Thank you very much for your valuable and meaningful suggestions related to the correction. In particular, I would like to express my deep appreciation for the suggestions regarding the theme change. We asked MDPI the correction of English, and corrected manuscript. Thank you for re-peering.

Kind regards

     Kiharu Igarashi

 March 03, 2022
